# Crystallographic observation of nonenzymatic RNA primer extension

Wen Zhang[1,2,3], Travis Walton[1,2,3], Li Li[1,2,3], Jack W Szostak[1,2,3]*

[1]Department of Molecular Biology, Howard Hughes Medical Institute, Massachusetts General Hospital, Boston, United States; [2]Department of Genetics, Harvard Medical School, Boston, United States; [3]Center for Computational and Integrative Biology, Massachusetts General Hospital, Boston, United States

**Abstract** The importance of genome replication has inspired detailed crystallographic studies of enzymatic DNA/RNA polymerization. In contrast, the mechanism of nonenzymatic polymerization is less well understood, despite its critical role in the origin of life. Here we report the direct observation of nonenzymatic RNA primer extension through time-resolved crystallography. We soaked crystals of an RNA primer-template-dGMP complex with guanosine-5′-phosphoro-2-aminoimidazolide for increasing times. At early times we see the activated ribonucleotides bound to the template, followed by formation of the imidazolium-bridged dinucleotide intermediate. At later times, we see a new phosphodiester bond forming between the primer and the incoming nucleotide. The intermediate is pre-organized because of the constraints of base-pairing with the template and hydrogen bonding between the imidazole amino group and both flanking phosphates. Our results provide atomic-resolution insight into the mechanism of nonenzymatic primer extension, and set the stage for further structural dissection and optimization of the RNA copying process.

DOI: https://doi.org/10.7554/eLife.36422.001

*For correspondence:
szostak@molbio.mgh.harvard.edu

**Competing interests:** The authors declare that no competing interests exist.

## Introduction

Many lines of evidence point to a central role for RNA in the emergence of life on Earth (*Sutherland, 2016*; *Wachowius et al., 2017*). As part of the RNA world hypothesis, the earliest forms of life must have replicated their simple RNA genomes without the aid of complex enzymes (*Orgel, 2004*). A key part of this process would have been the nonenzymatic template directed polymerization of activated RNA nucleotides. The resulting complementary strand could then act as a template to produce a copy of the original sequence. Previous studies have suggested that the local conformation at the reaction site greatly impacts the nonenzymatic RNA polymerization reaction. For example, a wide range of nucleic acid backbones, such as peptide (*Schmidt et al., 1997*) or threose nucleic acids (*Heuberger and Switzer, 2006*), can template the polymerization of activated RNA monomers, but the fastest reactions occur on A-form structured RNA or LNA templates (*Zielinski et al., 2000*; *Schrum et al., 2009*). These results indicate that the A-form conformation of the template provides a structure conducive for the polymerization reaction. Downstream helper oligomers, commonly utilized to improve the yield of primer extension reactions, are also thought to improve the reaction in part by stabilizing the A-form structure of an otherwise disordered ssRNA template (*Vogel et al., 2005*; *Prywes et al., 2016*). Structural aspects of the template near the reaction site may underlie many of the characteristics of nonenzymatic RNA polymerization, such as the rate and fidelity of polymerization (*Leu et al., 2013*). A better atomic-level understanding of the structural factors that promote nonenzymatic RNA polymerization could help to optimize the reaction for generalized copying of sequences, thus leading to a better understanding of the conditions required for RNA replication during the origin of life.

**eLife digest** Enzymes speed up chemical reactions that are essential to life. Most enzymes are proteins, but some are molecules of ribonucleic acid or RNA. Like DNA, RNA is made from a chain of building blocks called nucleotides. In modern organisms, protein-based enzymes build RNAs by linking nucleotides together, while the building blocks of proteins are linked by an RNA-based enzyme at the heart of a structure called a ribosome. The earliest life on Earth most likely relied only on RNA-based enzymes, but during the emergence of life, scientists believe that RNA molecules must have replicated spontaneously before dedicated RNA-based enzymes had evolved.

How RNA could replicate without enzymes has been a puzzle for decades. Recently, scientists discovered a previously unsuspected chemical intermediate that forms during the process, and hypothesized that this molecule's special structure is what enables the chemical reaction that adds new nucleotides to a growing strand of RNA.

To test this hypothesis, Zhang et al. diffused free RNA nucleotides into a crystalized complex containing template strands of RNA attached to short pieces of RNA called primers, which kick-start replication. Then, the crystals were frozen at various intervals and viewed using X-rays. This allowed Zhang et al. to observe the structural changes that occurred over time as the compounds reacted. The approach first revealed that the free nucleotides had paired with complementary nucleotides on the RNA template strands. Then, pairs of free nucleotides reacted with each other to form the intermediate. Finally, the intermediate reacted with the primer, forming a new bond that connects the RNA primer to one of the nucleotides of the intermediate, while the other nucleotide of the intermediate was released as a free nucleotide.

This experiment confirms that the specific structure of the intermediate molecule promotes RNA replication without help from enzymes. These findings will benefit chemists and biologists who study how RNA evolves and replicates. Future research building upon this work will deepen scientific understanding of the environmental conditions that were required for life to appear on Earth.
DOI: https://doi.org/10.7554/eLife.36422.002

Crystallography is a powerful tool for gaining atomic-level understanding of the mechanism of nonenzymatic RNA polymerization. However, the widely used nucleoside-5′-phosphoro-imidazolide substrates are too labile to use directly in crystallographic studies. We have instead utilized stable phosphonate analogs, co-crystallized with a primer template duplex, to help model the structure of the reaction site (*Zhang et al., 2016*; *Zhang et al., 2018*). These studies indicate that pseudo-activated guanosine monomers bind the template through both Watson-Crick and non-canonical modes of hydrogen bonding, with the phosphonate-leaving group mimic highly disordered. In contrast, the symmetrical 5′−5′ triphosphate-linked dinucleotide GpppG, which is an analogue of the imidazolium-bridged dinucleotide intermediate, was observed in a crystal structure to bind the template through two consecutive Watson-Crick base-pairs, with the 3′-hydroxyl of the primer positioned for in-line attack (*Zhang et al., 2017*). This observation suggested that the high reactivity of the imidazolium-bridged dinucleotide intermediate could be due to a favorable preorganized geometry in the template bound state. Consistent with this model, the superiority of the newly identified nucleotide-5′-phosphoro-2-aminoimidazolide (2-AIpN) monomers (*Li et al., 2017*), is thought to be due to the greater stability of the corresponding imidazolium-bridged dinucleotide intermediate (Np-AI-pN) (*Walton and Szostak, 2017*). In addition, our recent study of the mechanism by which downstream helper oligonucleotides enhance the rate of nonenzymatic primer extension (*Zhang et al., 2018*) showed that the helper oligomers pre-organize the reaction site and decrease the distance between the 3′-hydroxyl of the primer and the phosphate of the adjacent nucleotide.

To fully understand the mechanism of nonenzymatic template-directed primer extension, static 3D structures of RNA primer-template complexes co-crystallized with substrate and intermediate analogues, or even with the actual substrates and intermediates, are not sufficient to depict the dynamic progress of the reaction over time. A molecular movie of actual substrates reacting on the template, captured through the methods of crystallography, would provide greater insight. The approach of time-resolved X-ray crystallography has been greatly developed to study the dynamic processes of macromolecules and small molecules, by utilizing X-ray probe pulses in the pump-

probe scheme (*Ki et al., 2017*). The technique of time-resolved serial femtosecond crystallography, as well as the conventional Laue-type polychromatic time-resolved crystallography (*Ren et al., 1999*), has been widely applied to probe the structural transitions of proteins in both the solid crystal phase and in solution. Another category of time-resolved X-ray crystallography depends on the diffusion of small molecules into protein crystals. For the study of enzymes, it has been found that catalytic activity is sometimes preserved in crystallized proteins, which allows a structural analysis of the mechanism through time-resolved X-ray crystallography (*Hajdu et al., 2000*). In such experiments, the catalytic process can be triggered by diffusion-based reaction initiation techniques. At various time points after the initiation of the reaction, the crystal is placed in liquid nitrogen to freeze the reaction. This technique has provided deep insight into the polymerization reaction catalyzed by DNA polymerase η. In addition to capturing intermediate steps in phosphodiester bond formation, these structural studies provided evidence for a previously unknown third $Mg^{2+}$ ion that catalyzes the rate-limiting step of the reaction (*Nakamura et al., 2012*; *Gao and Yang, 2016*).

Inspired by the above studies, we conducted a time-resolved analysis of nonenzymatic RNA polymerization by soaking chemically activated substrates into crystals containing a primer-template duplex. Here, we present the crystal structures of the template-bound substrate 2-AIpG (*Figure 1A*) and the reaction intermediate Gp-AI-pG (*Figure 1B*). These structures corroborate our previous studies using stable analogs and uncover new features of the template-bound substrates. By obtaining crystal structures at various times after the initiation of the reaction, we were able to capture a sequence of molecular 'snapshots' that follow the progress of the nonenzymatic RNA polymerization reaction. This reaction sequence consists of monomers binding to the template in multiple conformations, followed by formation of the imidazolium-bridged dinucleotide intermediate, and finally new phosphodiester bond formation between the primer and the adjacent nucleotide of the intermediate. These experiments strongly suggest that the observed conformation of the template-bound dinucleotide intermediate Gp-AI-pG is pre-organized so as to favor the primer extension reaction. The approach described herein should be broadly useful for answering additional mechanistic questions regarding nonenzymatic RNA polymerization.

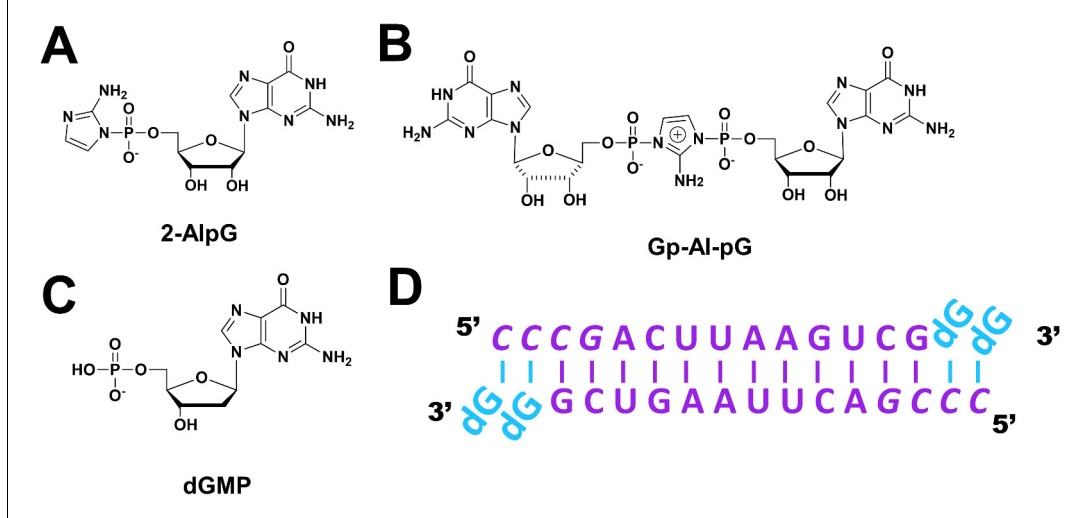

**Figure 1.** Chemical structures of mononucleotide and dinucleotide molecules in this study. (**A**) guanosine-5'-phosphoro-2-aminoimidazolide (2-AIpG). (**B**) 2-aminoimidazolium-bridged guanosine dinucleotide intermediate (Gp-AI-pG). (**C**) 2'-deoxyguanosine-5'-monophosphate (dGMP). (**D**) Schematic of the RNA-dGMP complex used for crystallographic studies. The four italic nucleotides at the 5'-end represent locked nucleic acid.
DOI: https://doi.org/10.7554/eLife.36422.003

## Results

### Structure of the primer-template duplex co-crystallized with dGMP

In order to follow nonenzymatic primer extension by crystallography, we first prepared crystals of an RNA primer-template complex in which unactivated monomers were bound to the template region. These crystals were used in crystal soaking experiments to exchange the unactivated monomers with activated monomers or the reaction intermediate. To facilitate comparison with previously obtained structures, we used the same partially self-complementary RNA 14-mer oligonucleotide that we used in our structural studies of the stable substrate analogues guanosine 5′-(3-methyl-1H-pyrazol-4-yl)phosphonate (PZG) and guanosine 5′-(4-methylimidazolyl) phosphonate (ICG), and the stable intermediate analogue GpppG (*Zhang et al., 2016*; *Zhang et al., 2017*; *Zhang et al., 2018*). Our attempts to crystallize a native RNA-monomer complex led to a highly disordered flanking template regions with no evidence for bound monomers. We therefore chose to use the oligoribonucleotide 5′-*CCCGACUUAAGUCG*-3′ which contains four locked nucleic acid (LNA) residues, denoted by italics. The LNA modification locks the sugar into the C3′-endo A-form conformation characteristic of a canonical RNA duplex. The two-nucleotide template overhang is composed of 5-methyl cytidine LNA to rigidify the backbone and facilitate crystallization. This duplex was co-crystallized with 2′-deoxyguanosine-5′-monophosphate (dGMP), to bind the template before soak-in experiments (*Figure 1C*). The use of dGMP allows unambiguous observation of the replacement of the initially bound deoxynucleotides with activated ribonucleotides during crystal soaking experiments, based upon the appearance of the electron density of the 2′-hydroxyl. Since the pyrazole and imidazole groups of monomer analogues are often disordered, the imidazole groups of the 2-AIpG would not be a reliable indicator of molecular exchange. Several other modified monomers, including 8-bromo-guanosine, guanosine-3′-monophosphate and guanosine 3′,5′-cyclic monophosphate, were tested for crystallization with the RNA duplex without success, possibly due to steric perturbation of molecular packing.

At both ends of the crystallized duplex, the two dGMP monomers bound to the -*CC* template in a well defined conformation (*Figure 2A*), exclusively through Watson-Crick base-pairing (*Figure 2B*). This result contrasts with our previous study of the RNA monomer analogues PZG and ICG, which displayed a variety of binding modes in addition to Watson-Crick base-pairing (*Zhang et al., 2016*; *Zhang et al., 2018*). The sugar and phosphate of the first bound dGMP (in the +1 position adjacent to the 3′ end of the primer) are well ordered, with the sugar in a C3′-endo A-form conformation. At the +2 position, the electron density clearly indicates the sugar is in the C3′-exo B-form conformation, but the phosphate is disordered. Previous crystal structures of DNA-RNA hybrid duplexes have shown DNA sugar puckers in both C3′-endo and C3′-exo conformations (*Horton and Finzel, 1996*). In contrast, our previous measurement using transferred nuclear Overhauser effect spectroscopy (TrNOESY) indicated that DNA monomers bound to an RNA template maintained a C2′-endo conformation (*Zhang et al., 2012*). This difference may result from the LNA template used for crystallization, sequence differences or crystal packing forces.

In the crystal structure of the RNA duplex with dGMP, we also clearly observed electron density near the N7 of the first template-bound dGMP that appears to represent a $Mg^{2+}$ ion based on its octahedrally coordinated waters (*Figure 2C*); $Mg^{2+}$ was present in the crystallization buffer. Two water molecules coordinated to the presumed $Mg^{2+}$ are located within hydrogen bonding distance of N7 and O6 of the primer guanosine (distances 2.8 Å and 2.6 Å). A third water molecule coordinated to the presumed $Mg^{2+}$ is within hydrogen bonding distance of O6 of the first bound monomer, and is also hydrogen bonded to a fourth water molecule that appears to be hydrogen bonded to N7 of the second bound monomer (distances 3.0 Å, 2.7 Å and 2.8 Å). These proposed $Mg^{2+}$-mediated electrostatic and hydrogen-bonding interactions bridge the two bound monomers and the primer, and thus potentially stabilize the template-bound structure of dGMP.

To confirm the proposed $Mg^{2+}$-guanosine interaction, we replaced $Mg^{2+}$ with $Sr^{2+}$ in our crystallization buffer, to increase the electron density of the bound metal cation. We obtained crystals of the same RNA duplex in 20 mM $Sr^{2+}$ with the RNA nucleotide guanosine-5′-monophosphate bound to the template region, with the same overall structure as the other complexes. In this structure, much stronger electron density was present near N7 and O6 of the first template-bound GMP, indicating the presence of $Sr^{2+}$ (*Figure 2—figure supplement 1*). This $Sr^{2+}$ ion binds in a similar manner as the proposed $Mg^{2+}$ ion observed in the structure of template-bound dGMP, supporting our

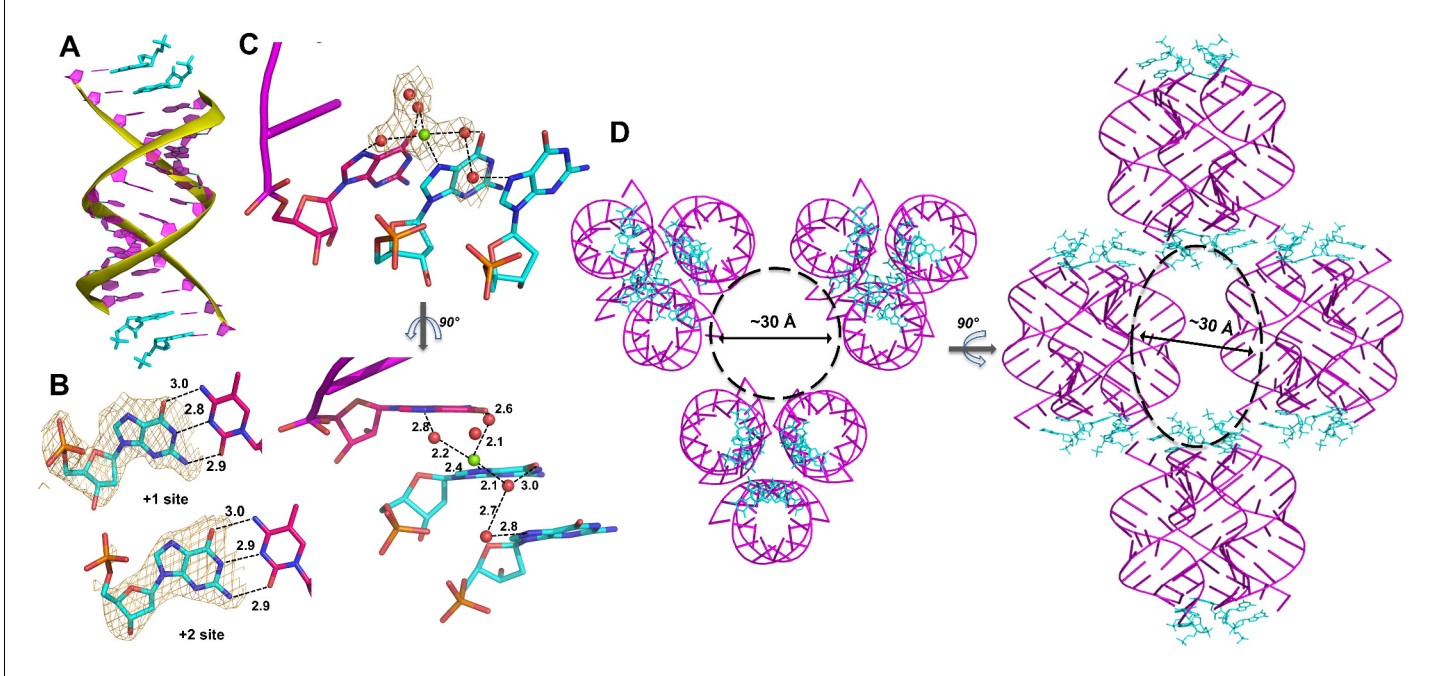

**Figure 2.** Crystal structure of RNA-dGMP complex. (A) One dimeric RNA duplex (purple with yellow backbone) is bound by two dGMP monomers (cyan) at each end. (B) dGMP monomers in the +1 and+2 position of the template form Watson-Crick base pairs. Wheat meshes indicate the corresponding $F_o$-$F_c$ omit maps contoured at 2.0 σ. (C) A $Mg^{2+}$ ion with coordinating water molecules form extended contacts between the 3′ end of the primer and both template-bound monomers through hydrogen bonds with N7. Wheat meshes indicate the corresponding $F_o$-$F_c$ omit maps contoured at 1.0 σ. (D) Top and side views of RNA-dGMP complex packing in crystal. Molecular packing of the RNA-dGMP complex shows a possible channel (dashed circle and oval) for diffusion of small molecules through the crystal.

DOI: https://doi.org/10.7554/eLife.36422.004

The following figure supplement is available for figure 2:

**Figure supplement 1.** A $Sr^{2+}$ ion and coordinating water molecule form an extended interaction network between the 3′ end of the primer and monomers in the +1 and+2 positions.

DOI: https://doi.org/10.7554/eLife.36422.005

identification of the interaction between N7 of dGMP in the +1 position with $Mg^{2+}$. Similar water-mediated metal-guanine interactions have previously been structurally identified in the deep major groove of A-form nucleic acid duplexes (*Robinson et al., 2000*; *Ennifar et al., 2003*).

Importantly, we observed a large channel in between the slip-stacked RNA duplexes that could allow activated substrates to reach the template by diffusion (*Figure 2D*). As in our previous studies, the self-complementary RNA duplex complexed with dGMP crystallized with hexagonal symmetry (space group P3$_1$21). The individual RNA-monomer double helices are slip-stacked with one another end-to-end, and groups of three adjacent duplexes form a triangular prism-like structure (*Figure 2D*, side view). Groups of three of these triangular prisms are in turn arranged in a triangular lattice, with a large central channel with a diameter of approximately 30 Å (*Figure 2D*, top view). This channel should allow mononucleotides or dinucleotides to diffuse into the crystal and bind the template in place of dGMP.

## Structure of the imidazolium-bridged intermediate bound to an RNA template

Having obtained crystals of the RNA-dGMP complex, we sought to replace the bound dGMP residues with purified imidazolium-bridged diguanosine intermediate (Gp-AI-pG). We soaked the crystals in buffer containing Gp-AI-pG for various times to allow the intermediate to diffuse into the crystal. No $Mg^{2+}$ was included in either the crystallization or soak buffers, to improve the solubility and stability of the Gp-AI-pG. The Gp-AI-pG molecule was observed by crystal structure determination to bind the template in place of dGMP, and optimal structures were obtained after 4 hr of

soaking the crystals in the exchange buffer. In the conditions used for crystal soaking, Gp-AI-pG decays with a half-life of 10 hr as observed by $^{31}$P NMR, indicating that the majority of Gp-AI-pG remains intact during this experiment (*Figure 3—figure supplement 1*). In the crystal structures, Gp-AI-pG binds the -CC template at both ends of the RNA duplex exclusively through Watson-Crick base-pairing (*Figure 3A*), with hydrogen bond distances of 2.8 to 3.3 Å. Similar to our previous analysis of the analog GpppG, the Gp-AI-pG intermediate is well-ordered and displays only the C3'-endo sugar conformation. The 4.6 Å distance between the 3'-hydroxyl and the phosphate of the nucleotide in the +1 position is shorter than the corresponding 6 to 6.5 Å distance for the 2'-

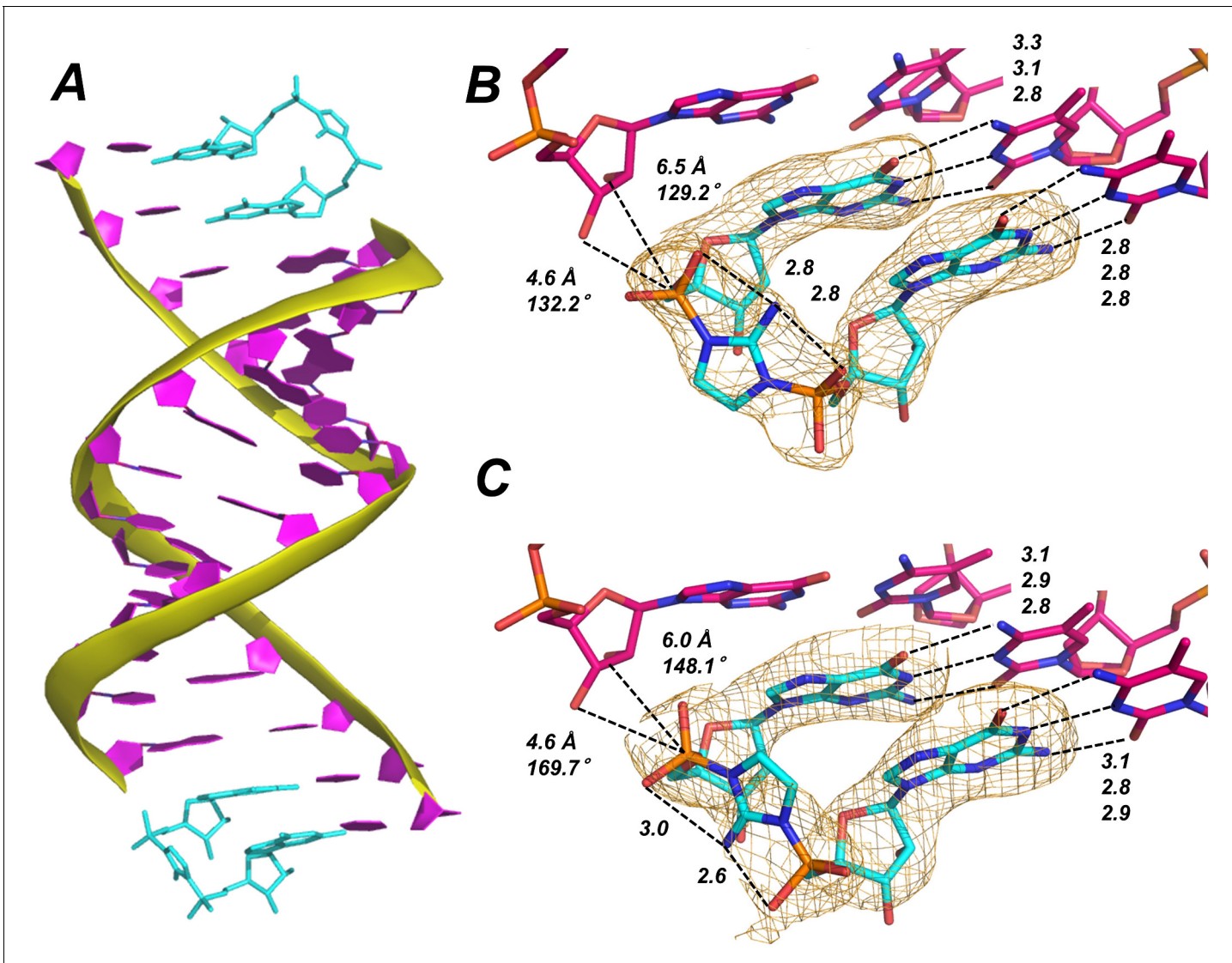

**Figure 3.** Template-bound Gp-AI-pG is pre-organized for phosphodiester bond formation. (A) Structure of RNA duplex (purple with yellow backbone) bound by two Gp-AI-pG molecules (cyan) obtained after 4 hr of soaking the RNA-dGMP complex in buffer containing Gp-AI-pG. (B) Close-up of template-bound Gp-AI-pG modeled with the 2-amino group of the imidazolium bridge pointing toward the major groove and (C) minor groove. Distances for hydrogen bonds and in-line attack by the primer 3'-hydroxyl are indicated by dashed lines. Wheat meshes indicate the corresponding $F_o$-$F_c$ omit maps contoured at 1.5 σ.

DOI: https://doi.org/10.7554/eLife.36422.006

The following figure supplement is available for figure 3:

**Figure supplement 1.** Decay of the Gp-AI-pG intermediate in crystal soaking buffer.

DOI: https://doi.org/10.7554/eLife.36422.007

hydroxyl, consistent with the observed 3′ regioselectivity of polymerization using 2-methyl and 2-aminoimidazole activated monomers (*Inoue and Orgel, 1981*; *Giurgiu et al., 2017*).

The two guanosine-5′-phosphate residues of Gp-AI-pG are linked by the 2-aminoimidazolium bridge. The electron density corresponding to the phosphate-aminoimidazole-phosphate bridge suggests that its structure is stabilized by hydrogen bonds between the 2-amino group of the imidazolium moiety and the non-bridging oxygen atoms of the flanking phosphates (*Figure 3B–C*). Unfortunately, the observed electron density does not clearly indicate whether the 2-amino group of the imidazolium bridge is pointing toward the major or minor groove of the duplex. The Gp-AI-pG intermediate is modeled in the two different orientations in *Figure 3B–C*. It may be that the two orientations are randomly distributed throughout the crystal. The orientation of the 2-aminoimidazolium group affects the distances and angle of attack between the primer 2′- or 3′-hydroxyl and the P-N bond to be broken during phosphodiester bond formation. For the 3′-hydroxyl, the distance to the phosphorous atom is 4.6 Å in both orientations, but angle of attack changes from 170° when the 2-amino points toward the minor groove, to 132° when it points towards the major groove. For the 2′-hydroxyl, the angle decreases from 148° to 129° but the distance increases from 6.0 to 6.5 Å when the 2-amino group points towards the major groove. These results suggest that rotation of the 2-aminoimidazolium group between the two phosphates may affect the rate and regioselectivity of nonenzymatic RNA polymerization.

A comparison between template-bound dGMP and Gp-AI-pG reveals similar overall structures (*Table 1*). A key difference is the decreased distance between the 3′-hydroxyl of the primer and the phosphate in the +1 position, from 6.5 Å for bound dGMP to 4.6 Å for template-bound Gp-AI-pG. This decreased distance is likely due to the constraint imposed by the imidazolium-bridge of Gp-AI-pG. No monovalent metal ions or water molecules are observed close to the N7 positions of Gp-AI-pG or the 3′ G of the primer.

## Structural visualization of nonenzymatic primer extension

Encouraged by our success at soaking the reactive Gp-AI-pG intermediate into preformed crystals of the RNA duplex, we asked whether soaking the activated monomer 2-AIpG into the crystal in the presence of $Mg^{2+}$ might lead to formation of the intermediate, possibly followed by reaction with the primer in the crystal. We therefore grew crystals of the RNA duplex in buffer at pH 7.0 containing dGMP and 20 mM $Mg^{2+}$. The RNA-dGMP crystals were then transferred to a new drop of the same pH 7.0 crystallization buffer, containing 20 mM 2-AIpG and 20 mM $Mg^{2+}$. Crystal soaking was terminated by equilibrating the crystal in cryo-protectant (35% 2-methyl-2,4-pentanediol, MPD) for 2 min followed by storage in liquid nitrogen prior to analysis. In the crystallization buffer, only 8% of the 2-AIpG decayed in 3 hr as determined by $^{31}P$ NMR, indicating that the majority of the monomer remains activated during these experiments (*Figure 4—figure supplement 1*).

After 5 min of soaking in exchange buffer containing 2-AIpG, the dGMP monomers co-crystallized with the RNA duplex had already been replaced by 2-AIpG monomers (*Figure 4A*). The rapid diffusion and exchange of the monomer 2-AIpG relative to the dimer Gp-AI-pG is clear from the

**Table 1.** Crystallographic and structural features of RNA-ligand complexes.

| | RNA-dGMP | RNA-Gp-AI-pG | Time-resolved structures (RNA-2-AIpG) | | | | | | |
| | | | Stage 1 (5 min) | Stage 2 (15 min) | Stage 3 (30 min) | Stage 4 (1 h) | Stage 5 (1.5 h) | Stage 6 (2 h) | Stage 7 (3 h) |
|---|---|---|---|---|---|---|---|---|---|
| PDB code | 6C8D | 6C8E | 6C8I | 6C8J | 6C8K | 6C8L | 6C8M | 6C8N | 6C8O |
| Ligand binding modes | Watson-Crick | Watson-Crick | mixed | Watson-Crick | Watson-Crick | Watson-Crick | Watson-Crick | Watson-Crick | Watson-Crick |
| N7 interaction | $Mg^{2+}$ | N.D. | $H_2O$ | $H_2O$ | N.D. | N.D. | N.D. | N.D. | N.D. |
| 3′-O-P distance, Å | ~6.1 | 4.6 | 3.7–4.1 | ~5.0 | ~4.3 | 4.1–5.0 | 1.6–4.2 | 1.6–4.1 | 1.6 |
| P-P distance between monomers, Å | N.D. | ~5.1 | N.D. | 6.2 | 5.2 | 5.2–5.4 | 5.5–7.1 | 5.0–7.3 | ~6.2 |
| N7-N7 distance between primer and + 1 monomer, Å | 4.2 | 3.7–4.0 | 3.9–4.1 | 4.0 | 3.7–4.0 | 3.7–4.2 | 3.7–4.3 | 3.7–4.1 | 4.1 |

N.D.: not detectable

DOI: https://doi.org/10.7554/eLife.36422.008

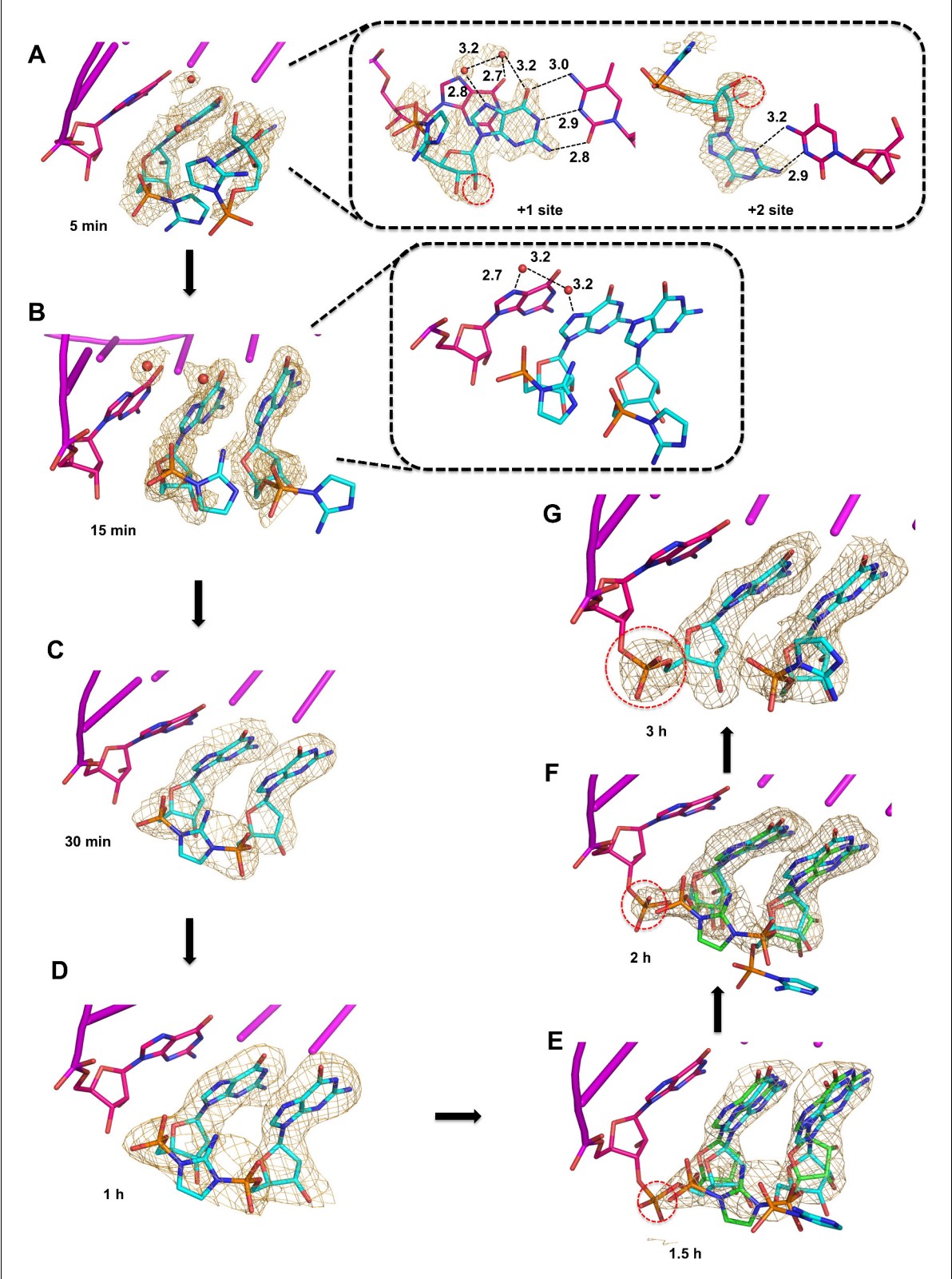

**Figure 4.** Observation of imidazolium-bridge and phosphodiester bond formation within crystals soaked with 2-AIpG. Wheat meshes in all structures indicate the corresponding $F_o$-$F_c$ omit maps contoured at 1.5 σ. (A) Structure after 5 min of soaking shows 2-AIpG monomer binding template through Watson-Crick and non-canonical base pairing. Inset also shows electron density belonging to 2′-hydroxyl (red dashed circle) and water molecules (red dots) that interact with N7 guanosine bases. (B) Structure after 15 min soaking shows both 2-AIpG monomers bound through Watson-Crick base

*Figure 4 continued on next page*

*Figure 4 continued*

pairing. Inset shows two water molecules that bridge the first monomer to the primer. (C) Structures after 30 min and (D) 1 hr of soaking shows the imidazolium-bridged Gp-AI-pG intermediate bound to the template. (E) Structures after 1.5 hr, and (F) 2 hr show a mix of template-bound Gp-AI-pG and extended primer (red dashed circle) (G) After 3 hr soaking, the electron density predominantly corresponds to the extension product.

DOI: https://doi.org/10.7554/eLife.36422.009

The following figure supplement is available for figure 4:

**Figure supplement 1.** Decay of the 2-AIpG monomer in crystal soaking buffer.

DOI: https://doi.org/10.7554/eLife.36422.010

appearance of the electron density of the 2'-hydroxyl in these structures, which cannot come from dGMP. The faster diffusion of the 2-AIpG monomer into the crystal is likely due to its smaller size. At this early time point, the 2-AIpG monomer at the +1 position is bound through Watson-Crick base pairing, but the monomer at the +2 position is bound through a non-canonical interaction. This atypical hydrogen bonding pattern has been observed in our previous studies of analogues for the guanosine-5'-phosphoro-2-methylimidazole monomer (*Zhang et al., 2016*). However, the 2-AIpG monomers in both the +1 and +2 positions are Watson-Crick base paired in the crystal structure obtained after 15 min of soaking the crystal (*Figure 4B*). These results further suggest that a variety of interactions are possible between monomers and the template during primer extension reactions. It is interesting to note that the distance between the primer 3'-hydroxyl and the 5'-phosphate of the incoming monomer increases when both monomers are bound to the template in Watson-Crick conformation (*Table 1*). In addition, we again observe electron density, likely corresponding to $Mg^{2+}$ and coordinated water molecules, that bridges from N7 of the 3'-G of the primer to N7 of the 2-AIpG monomer in the +1 position.

Structures obtained after continued incubation of the crystals with activated monomer 2-AIpG reveal the gradual appearance of the imidazolium-bridged dinucleotide Gp-AI-pG. While the 2-aminoimidazole groups are largely disordered and the imidazolium bridge is not observed in the structures obtained after 5 and 15 min of soaking, new density corresponding to formation of the imidazolium bridge clearly emerges after 30 min to 1 hr of soaking (*Figure 4C–D*). In addition, the ~6.2 Å (15 min) distance between the two phosphorus atoms of separate monomers decreases to 5.2 Å, very close to the 5.1 Å distance observed for the template-bound structure of Gp-AI-pG (*Table 1*). Similar to the structure obtained by soak-in of Gp-AI-pG (*Figure 3*), these structures show the dinucleotide intermediate bound to the template through two Watson-Crick base pairs and an imidazolium bridge that is ordered through hydrogen bonds between the 2-amino group and the non-bridging oxygen atoms of the phosphates. These results suggest that the 2-AIpG monomers are reacting inside the crystal to form the dinucleotide intermediate Gp-AI-pG.

When the crystal was incubated in a solution of 2-AIpG for times over 1 hr, we began to observe the appearance of a new phosphodiester bond between the primer and the adjacent G residue (*Figure 4E–G*). At times from 1 to 3 hr, new electron density appears between the primer 3'-hydroxyl and the 5'-phosphate of the monomer in the +1 position. The structures at 1.5 hr and 2 hr represent the average of a distribution of chemical states, and are refined as a mixture of template-bound Gp-AI-pG and the primer +1 extension product with a 2-AIpG monomer in the +2 binding site of the template. The ratio of these two states changes over time as phosphodiester bond formation proceeds to near completion by 3 hr. The structure at 3 hr clearly indicates a new well-ordered phosphodiester bond, with a 2-AIpG monomer bound to the template in the +2 position through Watson-Crick base pairing. The distance between the N7 atoms of the primer guanosine and the incoming guanosine remains the same throughout the reaction in the crystal (*Table 1*), suggesting that the base pairs and base stacking features of the RNA-substrate duplex are not perturbed during the course of the reaction. Only local conformational changes associated with formation and then loss of the imidazolium-bridge and formation of the new phosphodiester bond are observed. These results are consistent with a recent analysis indicating that the predominant mechanism of primer extension by 2-aminoimidazole activated monomers is through the imidazolium-bridged dinucleotide intermediate (*Walton and Szostak, 2017*). However, the observed decline in density for the imidazolium bridge in our current structures may also be due in part to hydrolysis of Gp-AI-pG.

## Discussion

The mechanism of template-directed nonenzymatic RNA polymerization has recently been shown to involve an imidazolium-bridged dinucleotide intermediate (*Walton and Szostak, 2016*; *Kervio et al., 2016*, also noted the high reactivity of this molecule). A major question is why the primer reacts so much more rapidly with the template-bound intermediate than with template-bound monomers. Our previous investigation of GpppG, a stable analog of the Gp-AI-pG intermediate, confirmed that a 5′,5′ linked dinucleotide could bind the template through two Watson-Crick base pairs and suggested that the true intermediate would be pre-organized for phosphodiester bond formation (*Zhang et al., 2017*). In contrast, the leaving group is highly disordered in our previously determined structures of stable phosphonate analogues of activated monomer substrates bound to the same template. Our current studies using the actual substrate 2-AIpG and intermediate Gp-AI-pG strongly support the model that structural pre-organization contributes to the reactivity of the covalent intermediate. The distance from the primer 3′-hydroxyl to the adjacent phosphate decreases from 6.5 Å for the monomer 2-AIpG, to 4.6 Å for the intermediate Gp-AI-pG. In addition, hydrogen bonding between the 2-amino group of the imidazolium bridge and the flanking non-bridging phosphate oxygen atoms likely helps to stabilize the conformation of the imidazolium bridge and pre-organize the template-bound intermediate for reaction with the primer. In contrast, the 2-aminoimidazole group of 2-AIpG was disordered in our structures, suggesting that the formation of a covalent link between two activated nucleotides is required to organize the correct conformation of the leaving group. We note that the hydrogen bonds of the 2-aminoimidazolium bridge would be absent for other imidazole groups such as 2-methylimidazole or 2-methylaminoimidazole. In addition, bulkier substitutions on the imidazole ring may distort the structure of the intermediate through steric clashes with the non-bridging oxygen atoms of the flanking phosphates. These considerations may help to explain the superiority of 2-aminoimidazole activation relative to other imidazole groups (*Li et al., 2017*).

Our observation of phosphodiester bond formation in the crystal strongly suggests the relevance of these structures for template-directed nonenzymatic RNA polymerization. However, the rate of the primer extension reaction is much slower in the crystal than in solution. In the crystal structure of template-bound Gp-AI-pG, the distance between the 3′-hydroxyl and the 5′-phosphate of the incoming nucleotide is quite long at ~4.6 Å. Clearly, significant conformational changes must occur before reaction with the primer, and these changes may be constrained by molecular packing. Such constraints may explain the slow rate of reaction in the crystal. In our recently determined structures in which GpppG is sandwiched between the primer and a downstream helper, the O3′-P distance is reduced, consistent with the enhanced reaction rate observed in the presence of a downstream helper oligonucleotide. If the Gp-AI-pG intermediate can be assembled in between the primer and a downstream oligonucleotide in crystal soaking experiments, and a decreased O3′-P distance is observed, primer extension in the crystal will require smaller conformational changes and may proceed more rapidly. The strategy of soaking reactive substrates into a pre-crystallized RNA duplex followed by time-resolved X-ray crystallography could also be applied to address other aspects of the reaction mechanism and outcome. For example, a mismatch at the +1 position results in decreased 3′−5′ regioselectivity, but the structural basis of this observation remains unclear (*Giurgiu et al., 2017*).

Many important aspects of phosphodiester bond formation between the primer and template-bound Gp-AI-pG remain unclear from our current series of time-resolved X-ray crystal structures. Foremost among these unanswered questions is the role of the catalytic metal ion in catalysis of the primer extension reaction. The primer extension reaction we observed in the crystal was $Mg^{2+}$ catalyzed, since the reaction was observed only when the monomer 2-AIpG was soaked into the crystal in the presence of $Mg^{2+}$. In contrast, when Gp-AI-pG was soaked into the crystal without $Mg^{2+}$, no reaction was observed. In our structures, the only $Mg^{2+}$ interactions observed involved bridging the N7 atoms of the primer and the monomer at the +1 position. These interactions between consecutive G nucleotides in the primer and template may help to properly orient the G monomers, and may help to explain the historically faster nonenzymatic template directed primer extension observed with G compared with the other nucleotides. However, $Mg^{2+}$ has long been thought to have a direct catalytic role through activation of the primer 3′-hydroxyl nucleophile and possibly also through interactions with the monomer phosphate. Although we did not observe any bound $Mg^{2+}$

near the site of phosphodiester formation, this is not surprising due to the very weak binding of the catalytic metal ion. Strategies to decrease the O3′-P distance and to further pre-organize the reaction center, for example through the use of a downstream helper oligonucleotide and/or the synthesis of transition state analogues, may allow visualization of the catalytic metal ion in future studies.

# Materials and methods

## Key resources table

| Reagent type (species) or resource | Designation | Source or reference | Identifiers | Additional information |
|---|---|---|---|---|
| Sequence-based reagent | synthetic RNA | Exiqon Inc. | | |
| Commercial kit | Crystallization Screening kits | Hampton research, Inc. | | |
| Software | Refmac5 | University of Cambridge DOI: 10.1107/S0907444996012255 | RRID:SCR_014225 | |
| Software | HKL2000 | HKL Research Inc. DOI: 10.1016/S0076-6879(97)76066-X | RRID:SCR_015547 | |
| Software | Phaser 2.7 | University of Cambridge DOI: 10.1107/S0021889807021206 | RRID:SCR_014219 | |
| Software | MestReNova | Mestrelab Research, Inc. | | |
| Software | Pymol2 | Schrödinger, Inc. | RRID:SCR_000305 | |

## Preparation of oligonucleotides, RNA crystallization, data collection, and structure determination

The oligonucleotide 5′-CCCGACUUAAGUCG-3′ was synthesized in-house by standard solid-phase techniques using native and locked nucleoside phosphoramidites from Exiqon Inc., followed by HPLC purification. C denotes LNA 5-methylcytidine residues and G denotes LNA guanosine residues, while A, C, G and U denote unmodified RNA residues. For the annealing step prior to crystallization, the RNA duplex (1 mM) was mixed with an equal volume of dGMP (50 mM) and heated to 80° C for 2 min before being slowly cooled to room temperature. All crystals were grown at 18° C. The Nucleic Acid Mini Screen Kit, Natrix High Throughput Kit and Index High Throughput Kit (Hampton Research, Aliso Viejo, CA) were used for screening crystallization conditions by the hanging drop or sitting drop vapor diffusion methods. Optimal crystals grew in a crystallization buffer of 5% v/v (+/-)−2-methyl-2, 4-pentanediol, 20 mM sodium cacodylate pH 7.0, 6 mM spermine tetrahydrochloride, 40 mM sodium chloride, and 20 mM magnesium chloride. Crystals used for soak-in of Gp-AI-pG were obtained under identical conditions except that magnesium chloride was omitted. For crystallization of the RNA duplex with GMP, 20 mM $SrCl_2$ was added in place of $MgCl_2$. To soak 2-AIpG or Gp-AI-pG into crystals of the RNA duplex with dGMP, the preformed crystals were first stabilized in a drop containing crystallization buffer for 20 min. Then, the crystals were transferred to a drop containing crystallization buffer containing 20 mM Gp-AI-pG or 2-AIpG and incubated for various amounts of time to allow diffusion of the molecules into the crystal. As with the crystallization procedure, $Mg^{2+}$ was omitted for soaking Gp-AI-pG into the crystal. After incubation with the activated substrates, the crystal was dipped into cryo-protectant (35% MPD) for 2 min and then placed in liquid nitrogen.

All crystal diffraction data were collected under a stream of nitrogen at 99 K. The data sets were obtained at the SIBYLS beam lines 821 and 822 at the Advanced Light Source, Lawrence Berkeley National Laboratory. The distance from the detector to the crystal was set between 200–300 mm, and the collecting wavelength was set to 1 Å. The crystals were exposed for 1 s per image with one degree oscillations, and 180 images were taken for each data set. These data were processed using HKL2000 (*Otwinowski and Minor, 1997*). All the structures were solved by molecular replacement (*McCoy et al., 2007*). The RNA models were from our previously reported structure (PDB code 5DHC). All the structures were refined using Refmac (*Murshudov et al., 1997*). The refinement protocol included simulated annealing, refinement, restrained B-factor refinement, and bulk solvent correction. During refinement, the topologies and parameters for locked nucleic acids and for the

ligand 2-AIpG or Gp-AI-pG were constructed and applied. After several cycles of refinement, the water and metal atoms were added. Data collection, phasing, and refinement statistics of the determined structures are listed in *Supplementary file 1*.

The structures at 1.5 and 2 hr were refined as mixtures of the intermediate state and product state.

## Preparation of activated substrates 2-AIpG and Gp-AI-pG and analysis of decay

2-AIpG and Gp-AI-pG were synthesized and purified as previously described (*Li et al., 2017*; *Zhang et al., 2017*). To characterize the stability of these substrates during the crystal soaking procedure, samples of 2-AIpG and Gp-AI-pG were prepared in crystallization buffer and monitored by $^{31}P$ NMR. These samples were placed in a Shigemi tube with a co-axial insert containing $D_2O$ for locking the NMR signal. All NMR spectra were taken on a Varian INOVA 400 MHz NMR spectrometer at 25°C and analyzed by integration of peaks using MestReNova software. $^{31}P$ NMR signals are referenced to internal trimethyl phosphate added after the experiments. The NMR spectra were shown in *Figure 3—figure supplement 1* and *Figure 4—figure supplement 1*.

## Acknowledgements

We thank Dr. Derek O'Flaherty for insightful comments on the manuscript. We thank all members of the Szostak lab for helpful discussions, and Dr. Shui-Ying Ng for operational assistance. We also thank the staff at the Advanced Light Source (ALS) SIBYLS beamlines 8.2.1 and 8.2.2, a national user facility operated by Lawrence Berkeley National Laboratory on behalf of the Department of Energy, Office of Basic Energy Sciences, through the Integrated Diffraction Analysis Technologies (IDAT) program, supported by the DOE Office of Biological and Environmental Research. JWS is an Investigator of the Howard Hughes Medical Institute. This work was supported in part by grants from the National Science Foundation [CHE-1607034] and the Simons Foundation [290363] to JWS. L L is a Life Sciences Research Foundation fellow. Funding for open access charge: Howard Hughes Medical Institute and the Simons Collaboration on the Origins of Life.

## Additional information

### Funding

| Funder | Grant reference number | Author |
|---|---|---|
| Howard Hughes Medical Institute | | Jack W Szostak |
| National Science Foundation | CHE-1607034 | Jack W Szostak |
| Simons Foundation | 290363 | Jack W Szostak |

The funders had no role in study design, data collection and interpretation, or the decision to submit the work for publication.

### Author contributions

Wen Zhang, Supervision, Writing—original draft, Writing—review and editing; Travis Walton, Li Li, Jack W Szostak, Writing—original draft, Project administration, Writing—review and editing

### Author ORCIDs

Wen Zhang http://orcid.org/0000-0003-4811-4384
Travis Walton http://orcid.org/0000-0001-6812-1579
Li Li http://orcid.org/0000-0003-4766-5782
Jack W Szostak http://orcid.org/0000-0003-4131-1203

### Decision letter and Author response

Decision letter https://doi.org/10.7554/eLife.36422.034

Author response https://doi.org/10.7554/eLife.36422.035

## Additional files

### Supplementary files
• Supplementary file 1. Table S1: Data collection statistics. Table S2: Structure refinement statistics.
DOI: https://doi.org/10.7554/eLife.36422.011
• Transparent reporting form
DOI: https://doi.org/10.7554/eLife.36422.012

### Data availability
Diffraction data have been deposited in PDB under the accession code 6C8D, 6C8E, 6C8I, 6C8J, 6C8K, 6C8L, 6C8M, 6C8N, 6C8O, 6CAB.

The following datasets were generated:

| Author(s) | Year | Dataset title | Dataset URL | Database, license, and accessibility information |
|---|---|---|---|---|
| Zhang W, Szostak JW | 2018 | RNA-dGMP complex with Mg ion | http://www.rcsb.org/pdb/search/structid-Search.do?structureId=6C8D | Publicly available at the RCSB Protein Data Bank (accession no: 6C8D) |
| Zhang W, Szostak JW | 2018 | RNA-imidazolium-bridged intermediate complex, 4h soaking | http://www.rcsb.org/pdb/search/structid-Search.do?structureId=6C8E | Publicly available at the RCSB Protein Data Bank (accession no: 6C8E) |
| Zhang W, Szostak JW | 2018 | RNA-activated 2-AIpG monomer complex, 5 min soaking | http://www.rcsb.org/pdb/search/structid-Search.do?structureId=6C8I | Publicly available at the RCSB Protein Data Bank (accession no: 6C8I) |
| Zhang W, Szostak JW | 2018 | RNA-activated 2-AIpG monomer complex, 15 min soaking | http://www.rcsb.org/pdb/search/structid-Search.do?structureId=6C8J | Publicly available at the RCSB Protein Data Bank (accession no: 6C8J) |
| Zhang W, Szostak JW | 2018 | RNA-activated 2-AIpG monomer complex, 30 min soaking | http://www.rcsb.org/pdb/search/structid-Search.do?structureId=6C8K | Publicly available at the RCSB Protein Data Bank (accession no: 6C8K) |
| Zhang W, Szostak JW | 2018 | RNA-activated 2-AIpG monomer complex, 1h soaking | http://www.rcsb.org/pdb/search/structid-Search.do?structureId=6C8L | Publicly available at the RCSB Protein Data Bank (accession no: 6C8L) |
| Zhang W, Szostak JW | 2018 | RNA-activated 2-AIpG monomer, 1.5h soaking | http://www.rcsb.org/pdb/search/structid-Search.do?structureId=6C8M | Publicly available at the RCSB Protein Data Bank (accession no: 6C8M) |
| Zhang W, Szostak JW | 2018 | RNA-activated 2-AIpG monomer complex, 2h soaking | http://www.rcsb.org/pdb/search/structid-Search.do?structureId=6C8N | Publicly available at the RCSB Protein Data Bank (accession no: 6C8N) |
| Zhang W, Szostak JW | 2018 | RNA-activated 2-AIpG monomer, 3h soaking | http://www.rcsb.org/pdb/search/structid-Search.do?structureId=6C8O | Publicly available at the RCSB Protein Data Bank (accession no: 6C8O) |
| Zhang W, Szostak JW | 2018 | RNA-dGMP complex with Sr2+ ion | http://www.rcsb.org/pdb/search/structid-Search.do?structureId=6CAB | Publicly available at the RCSB Protein Data Bank (accession no: 6CAB) |

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
