## [Decision Letter]

Thank you for submitting your article "Crystallographic observation of non-enzymatic RNA primer extension" for consideration by *eLife*. Your article has been reviewed by three peer reviewers, and the evaluation has been overseen by Timothy Nilsen as Reviewing Editor and Michael Marletta as the Senior Editor. The following individuals involved in review of your submission have agreed to reveal their identity: Gerald F Joyce (Reviewer #1); Steven Benner (Reviewer #2); John Chaput (Reviewer #3).

The reviewers have discussed the reviews with one another and the Reviewing Editor has drafted this decision to help you prepare a revised submission.

Summary:

Zhang et al. have obtained a series of X-ray crystal structures that capture the reaction pathway for non-enzymatic, RNA-templated addition of an activated mononucleotide to the 3' end of an RNA primer.

All of the reviewers agreed that the manuscript presented exciting and important work. They further agreed that only minor revision was required before the work could be accepted by *eLife*. Please address these points as thoroughly as possible before resubmitting.

Reviewer #1:

Zhang et al. have obtained a series of X-ray crystal structures that capture the reaction pathway for non-enzymatic, RNA-templated addition of an activated mononucleotide to the 3' end of an RNA primer. The activating group is the long-studied 5'-phosphoroimidazolide, which only recently was shown to form an imidazolium-bridged dinucleotide that is the true reactive intermediate. The manuscript includes the structures of a non-reactive template-bound monomer (dGMP), a pre-reactive template-bound dimer (imidazolium-bridged diguanylate), and, most significantly, a time series of complexes that progress from the reactive template-bound monomer, through the reactive intermediate, to the extension product. "Seeing" this reaction for the first time brings chills. Even if the 2-aminoimidizole activating group is not literally what was employed at the dawn of the RNA world, visualizing the non-enzymatic copying of genetic information is a major accomplishment. This manuscript is highly suitable for publication in *eLife*, subject to minor revision.

1) It would be helpful to include in Figure 1 the paired structure of the template-primer complex, shown as a homodimer and indicating the sites on both strands for binding the two guanylate residues.

2) Clearly it would have been preferable if the template were all-RNA, rather than containing LNA residues. The authors state that this was done "to rigidify the backbone and facilitate crystallization". I presume that attempts were made to use all-RNA molecules, in which case this should be stated explicitly. It also should be noted that the LNA residues are "locked" into the C3'-endo conformation, which helps enforce an A-form duplex structure. Including this information at the outset will prepare the reader for the discussion in the following paragraph.

3) The authors state: "Surprisingly, no metal ion or water molecules are observed close to the N7 positions". My understanding is that no Mg^2+^ was included in the crystallization or soak buffer for these experiments, so at least with regard to metal ions it is not surprising they were not observed.

4) Figure 4 is simply "Wow!"; the payoff of decades of research. To gild the picture just a bit, it should be possible to make a quantitative estimate of the fraction of imidazolium-bridged diguanylate vs. extension product for the 1, 1.5, 2, and 3 hour time points. This assumes one can fit the electron density to a two-state model that corresponds to the fraction reacted at each time. These data in turn could be used to estimate the reaction rate within the crystal. As the authors note, this rate is much slower than in solution, which may be due to constraints of crystal packing, but could also be due to access to the catalytic Mg^2+^, attaining the proper solvation state, or other factors. The fact that the reaction is slower in the crystal does not diminish the significance of the results, and indeed was helpful for visualizing the reaction trajectory. Clearly that trajectory is qualitatively the same as for the reaction in solution.

Reviewer #2:

The problem that this manuscript addresses, the uncatalyzed template-directed synthesis of RNA, is one of the six or seven central problems in the origins of life.

Further, the result that the research reported in this manuscript has achieved, allowing crystallography to follow dynamic processes, here chemical reactions, has long been a "Holy Grail" goal of crystallography, dating back to the early 1980s when cryo-enzymology achieve this for a few cases, notably ribonuclease in the laboratory of Gregory Petsko. This work is evidently long forgotten. More recent work on time resolved x-ray crystallography is reviewed in the Introduction of this manuscript.

Therefore, this manuscript has chosen well both the subject for study and the approach of the study. I have no major criticisms that need response.

It is interesting that when one actually uses the true substrate, as opposed to a stable analog, organized electron density can be seen, contrasting with the disorder that was seen with the substrate analog. This is appropriately mentioned in the Discussion, but I don't know of any other case where this is been seen.

Reviewer #3:

This is an outstanding paper that describes a critical step in the origin of life. Using time-resolved X-ray crystallography, the authors follow the progress of a model non-enzymatic primer extension reaction where an activated ribonucleotide is added to the 3' end of an RNA primer. The reaction consists of monomers binding to the template in multiple conformations, followed by the formation of an imidazolium-bridged dinucleotide intermediate, and finally the formation of a new phosphodiester bond between the primer and adjacent monomer of the activated intermediate. The results strongly suggest that non-enzymatic primer extension reactions proceed by a mechanism where the activated imidazolium-ion intermediate is pre-organized on the template, which helps explain the high rate of synthesis of these RNA analogues. The manuscript was well written, and the experiments are clearly described in the main text and methods sections. Overall, I have no minor or major concerns and strongly feel that this paper would be excellent contribution to *eLife*.

---

## [Author Response]

Reviewer #1:[…] 1) It would be helpful to include in Figure 1 the paired structure of the template-primer complex, shown as a homodimer and indicating the sites on both strands for binding the two guanylate residues.

As requested, we have added a diagram to Figure 1 to illustrate the RNA sequence for all of the structural studies in our paper.

2) Clearly it would have been preferable if the template were all-RNA, rather than containing LNA residues. The authors state that this was done "to rigidify the backbone and facilitate crystallization". I presume that attempts were made to use all-RNA molecules, in which case this should be stated explicitly. It also should be noted that the LNA residues are "locked" into the C3'-endo conformation, which helps enforce an A-form duplex structure. Including this information at the outset will prepare the reader for the discussion in the following paragraph.

The reviewer is correct, in that we had indeed tried to co-crystallize the unmodified RNA template with monomers. Although we did obtain crystals, the solved structure only showed the central RNA duplex. The flanking template region was highly disordered, without any interpretable evidence for bound monomer. As a result we chose to use LNA in the template region to rigidify the structure. We have added a brief explanation in the first paragraph of the subsection “Structure of the primer-template duplex co-crystallized with dGMP”.

3) The authors state: "Surprisingly, no metal ion or water molecules are observed close to the N7 positions". My understanding is that no Mg^2+^ was included in the crystallization or soak buffer for these experiments, so at least with regard to metal ions it is not surprising they were not observed.

The reviewer is correct in that no Mg^2+^ was present in the crystallization buffer. We have deleted the word “surprisingly”, and modified the sentence to clarify that no monovalent ions or water molecules were observed close to N7.

4) Figure 4 is simply "Wow!"; the payoff of decades of research. To gild the picture just a bit, it should be possible to make a quantitative estimate of the fraction of imidazolium-bridged diguanylate vs. extension product for the 1, 1.5, 2, and 3 hour time points. This assumes one can fit the electron density to a two-state model that corresponds to the fraction reacted at each time. These data in turn could be used to estimate the reaction rate within the crystal. As the authors note, this rate is much slower than in solution, which may be due to constraints of crystal packing, but could also be due to access to the catalytic Mg^2+^, attaining the proper solvation state, or other factors. The fact that the reaction is slower in the crystal does not diminish the significance of the results, and indeed was helpful for visualizing the reaction trajectory. Clearly that trajectory is qualitatively the same as for the reaction in solution.

This is a point that we discussed at great length during the preparation of our original manuscript. Ultimately we decided against including such a plot, for fear that this would be an over-interpretation of the available data. The problem is that our refined models of the imidazolium-bridged dimer and the extension product do not account for 100% of the molecules in the crystal. It is likely that hydrolysis products such as GMP and 2-AIpG occupy some fraction of the template molecules. As a result, we do not feel able to quantitatively estimate the fraction of imidazolium-bridge vs. phosphodiester bond based on our observed electron densities. An additional confounding factor is that the rate of molecular diffusion into the crystal depends on the size the crystal, so quantitative conclusions based on structures from different crystals might be misleading. We certainly hope to overcome these limitations in future work, but believe that a simple qualitative characterization is most appropriate at present.

Reviewer #2:The problem that this manuscript addresses, the uncatalyzed template-directed synthesis of RNA, is one of the six or seven central problems in the origins of life.Further, the result that the research reported in this manuscript has achieved, allowing crystallography to follow dynamic processes, here chemical reactions, has long been a "Holy Grail" goal of crystallography, dating back to the early 1980s when cryo-enzymology achieve this for a few cases, notably ribonuclease in the laboratory of Gregory Petsko. This work is evidently long forgotten. More recent work on time resolved x-ray crystallography is reviewed in the Introduction of this manuscript.Therefore, this manuscript has chosen well both the subject for study and the approach of the study. I have no major criticisms that need response.It is interesting that when one actually uses the true substrate, as opposed to a stable analog, organized electron density can be seen, contrasting with the disorder that was seen with the substrate analog. This is appropriately mentioned in the Discussion, but I don't know of any other case where this is been seen.

We thank the reviewer for this appreciation of our work. We note that, as stated in the text (subsection “Structural visualization of non-enzymatic primer extension”, second paragraph), the 2-aminoimidazole leaving groups of the bound monomers are disordered – this is similar to what we previously observed with non-hydrolyzable analogs.

Reviewer #3:This is an outstanding paper that describes a critical step in the origin of life. Using time-resolved X-ray crystallography, the authors follow the progress of a model non-enzymatic primer extension reaction where an activated ribonucleotide is added to the 3' end of an RNA primer. The reaction consists of monomers binding to the template in multiple conformations, followed by the formation of an imidazolium-bridged dinucleotide intermediate, and finally the formation of a new phosphodiester bond between the primer and adjacent monomer of the activated intermediate. The results strongly suggest that non-enzymatic primer extension reactions proceed by a mechanism where the activated imidazolium-ion intermediate is pre-organized on the template, which helps explain the high rate of synthesis of these RNA analogues. The manuscript was well written, and the experiments are clearly described in the main text and methods sections. Overall, I have no minor or major concerns and strongly feel that this paper would be excellent contribution to eLife.

We thank the reviewer for this appreciation of our work.